# Non-Clinical Contribution to Clinical Trials during Lead Optimization Phase

**DOI:** 10.3390/bs8010017

**Published:** 2018-01-24

**Authors:** Lázara Martínez Muñoz

**Affiliations:** Cuban Council of Veterinary Sciences, Paseo 604 Vedado CP, La Habana 10400, Cuba; lazaramartinez@infomed.sld.cu; Tel.: +53-7-2714636

**Keywords:** non-clinical, translational medicine, first-in-human drug use

## Abstract

This manuscript comments on guidelines related to requirements for clinical trials for new drugs and the importance of considering regulatory criteria in the planning phase, in order to enhance the utility of data generated in basic research. Suggestions are made for optimizing regulatory management to improve the likelihood of acceptance of pre-clinical data prior to Clinical Phase I trials (early clinical trials).

## 1. Introduction

The growing incidence of neurodegenerative diseases makes it imperative for scientists conducting research in this field to design projects in ways that minimize delay for patients in need of access to new technologies or medicines. From a regulatory perspective, there is concern about the limited understanding many investigators have of the importance of correctly integrating regulatory approaches and criteria into their preliminary work; such understanding allows planning for an effective research program in new drug development that improves the chances of positive regulatory review in order to support progress to clinical trials. Another concern is the need of appreciating the importance of laboratory animal science as an important component of research project development—necessary from the outset when investigating a new medication.

The choice of a drug candidate is a progressive process [1,2] that requires integration of professionals from disciplines including chemistry, biochemistry, pharmacy, toxicology, and clinical medicine, among others. These professionals may be affiliated with different institutions that have diverse areas of focus and mission, yet must coordinate efforts. The understanding and inclusion of quality assurance programs in each experiment is an important aspect in keeping any data generated acceptable for regulatory approvals. At some institutions, the most-desired result may simply be the publication of data; however, that data may be relevant for supporting early clinical trials if the information is produced following appropriate criteria for quality assurance and meets regulatory requirements.

This manuscript reflects the presentation delivered during the last session of the congress “Non-Clinical Models for Neurodegenerative Diseases”, which was held in Cuba on 21–24 June 2017. Researchers representing different disciplines involved with the investigation of neurodegenerative diseases, ranging from those performing basic research to those conducting clinical trials, were invited congress participants. The congress was convened with the intention of facilitating a practical multidisciplinary exchange. The objective was achieved: participants appreciated the ability of dialogue between disciplines to streamline the development of medicines from laboratory to patient, as well as the important ways that organizational changes in basic research laboratories can facilitate acceptance by regulatory oversight bodies of the data generated.

## 2. International Guidelines Establish Data That Should Be Generated During the Investigative Phase

During the last decades, new concepts in clinical research have been introduced that allow for the clinical investigation of specific questions that rely on patients as well as healthy volunteers. These clinical trial options include Clinical Phase 0, micro-dosing studies, and adaptive designs for the selection of clinical doses [3,4], which can be performed with more than one candidate drug. Candidate drugs must have passed non-clinical studies specifically designed to generate confidence in the safety of the medication for people who will be treated, and to support the specific therapy that the clinical trial will examine. These pre-clinical studies are carried out as part of the initial research stage, prior to the regulated non-clinical phase and are key to deciding “yes or no”—whether the product continues to be investigated. Such a strategy helps reduce resource costs for non-optimal drug candidates, and speeds the process of developing new medicines.

The following regulatory guidelines offer examples of data required for documentation that is normally generated during the basic research and candidate optimization phase of development:Concept paper on the development of a Committee for Medicinal Products for Human Use (CHMP) guideline on the non-clinical requirements to support early Phase I clinical trials with pharmaceutical compounds [5]ICH Topic M 3 (R2) Non-Clinical Safety Studies for the Conduct of Human Clinical Trials and Marketing Authorization for Pharmaceuticals [6]Guideline on strategies to identify and mitigate risks for first-in human clinical trials with medical research products [7]Guideline on strategies to identify and mitigate risks for first-in-human and early clinical trials with medical research products (R1) [8]Guidance for industry, researchers and reviewers: Exploratory IND studies [9]Position paper on non-clinical safety studies to support clinical trials with a single micro-dose [10].

Concept Paper on the development of a CHMP Guideline on the non-clinical requirements to support early Phase I Clinical trials with pharmaceutical compounds [5] describes the need for additional guidelines useful to delineate requirements for early clinical trials while maintaining the protection of human research subjects, but which also allow sponsors to more efficiently advance the development of promising drug candidates. The result was the review “ICH M3: Non-Clinical Safety Studies for the Conduct of Human Clinical Trials and Marketing Authorization for Pharmaceuticals”. In March of 2009, the ICH Topic M 3 (R2) review was presented. Non-Clinical Safety Studies for the Conduct of Human Clinical Trials and Marketing Authorization for Pharmaceuticals [6] in section “Recommended Non-Clinical Studies to Support Exploratory Clinical Trial” describes the non-clinical requirements for the different variants of exploratory clinical trials. All of these documents establish as a common requirement that “In vitro target/receiver profiling should be conducted” and “Appropriate characterization of primary pharmacology (mode of action and/or effects) in a pharmacodynamically relevant model should be available to support human dose selection”.

The requirements may seem clear and simple, but when taken together, the scope of the work that is required to follow the requirements can be viewed as a guideline for strategies to identify and mitigate risks for the first-in-human clinical trials of investigative medicines [7]. The regulation is intended to direct sponsors and researchers during the transition to the early clinical trial of a research drug. Data for the evaluation of risk are generated in the research and optimization phase of candidate drug investigations, and the data need to comply with regulatory specifications in order to be useful in the evaluation and mitigation of risks to the human subjects. The absence of any relevant data results in a significant delay in moving on to the clinical stage of investigation.

This guide [7] identifies three risk factors: mode of action, nature of the target, and relevance of the animal species. Aspects such as the intensity of biological activity, possible interactions with other targets, activation and inhibition of other metabolic pathways, compensatory metabolic responses, and target polymorphisms, among other factors, need to be explored. Differences between humans and animal models should be investigated, and their relevance for dose calculation discussed. These differences between species include the distribution and affinity of targets to substances, consequences in cells and cellular regulatory mechanisms, and metabolic pathways.

In July of 2017 [8], a revision of the 2007 regulation [7] was released; the revision more clearly establishes the responsibility of sponsors and researchers to identify potential risks from the candidate drug prior to initiating a clinical study, and the measures to detect the risk and control unwanted effects. The revision emphasizes aspects related to the quality of the candidate drug, characterization of the targets, and mode of action, as well as specifications of animal models in relation to pharmacodynamic studies. Each of the risk factors identified in the 2007 guide are treated more exhaustively in the revised regulation—in particular, aspects related to animal models.

This regulation [8] also insists on the relevance of the assay selected to identify and evaluate the main pharmacological activity of the candidate drug. This aspect of relevance is of crucial importance. The assay should be able to quantitatively measure the response to the candidate drug, its potency, as well as the duration of the effect in both experimental models and in samples from humans [8,9,11]. The method chosen must have reliability and be related to the mechanism of action of the substance being evaluated. The assay is usually part of the trials related to the quality specifications of the candidate drug(s) in the clinical study [8,9].

An overview of these regulations [8,9] shows that in order to achieve expected results for the evaluation of potential risks, the molecular biochemists, physiologists, pharmacologists, chemists, pharmacists, and specialists in laboratory animal science must work collaboratively. Reliability is needed for tests to measure concentrations of product in the samples and for tests to assess power; consistency is required in results provided from both evaluation systems. The development of the test agent formulation also requires the previously mentioned data; in turn, evaluation of the efficacy of the formulations requires pharmacokinetic and pharmacodynamic assays in vitro and in vivo. Next, selection of the in vivo model requires that a laboratory animal science specialist have access to data on the test agent, and that this person is an integral part of the research team.

An efficient research development project requires these multidisciplinary team interactions, and the research results are only useful if principles of good practices are applied [12,13,14,15] and if the compilation of data is in accordance with regulatory specifications [16]. For test agents destined to be used in the diagnosis and treatment of neurodegenerative diseases, the selection of an animal model poses additional challenges. These challenges include the specificity of the human expression of many neurodegenerative diseases, the complexity of the mechanisms that generate the symptoms, as well as a lack of understanding of many pathophysiological processes involved in various neurological conditions. In addition, there are limitations to animal models, which may not express the broad spectrum of human symptoms. Finally, there is the difficulty posed by physiological and anatomical barriers that limit the access of test agents to penetrate the central nervous system.

## 3. Non-Clinical Safety Studies Have Different Strategies for Early Clinical Trials

FDA Guidance for Industry, Investigators, and Reviewers. Exploratory IND Studies. January 2006 [9]. Position paper on non-clinical safety studies to support clinical trials with a single microdose [10] and M3 (R2) Non-Clinical Safety Studies for the Conduct of Human Clinical Trials and Marketing Authorization for Pharmaceuticals in approach [6] and European Medicines Agency (EMEA). This guideline on strategies to identify and mitigate risks for first-in-human and early clinical trials with investigational medicinal products (R1) of 2017 [8] with few differences makes similar suggestions for safety studies for clinical trials prior to Clinical Phase I. Highlighted in these regulations are an extended set of parameters to be evaluated, as well as the consideration that an exaggerated pharmacological effect may be a cause of toxic effect, which should be included as a measure of the margin of safety. The extent of the studies and additional evaluation points depend on the characteristics of the investigational drug, the target, the disease to be treated, and the duration and treatment scheme in the exploratory clinical study. A single-dose mammalian toxicology study must be undertaken in both sexes. Route of test substance administration should be via the intended clinical route. Animals should be observed for 14 days post-dose. Necropsy should be performed at 24 h, 2 days, and 14 days post-dosing. Endpoints should include—at a minimum—body weights, clinical signs, clinical chemistries, hematology, and histopathology.

## 4. Selection of Doses for Exploratory Clinical Trials

The selection of doses for the first-in-human use of a candidate drug is based on previously collected data, much of which is generated in the research phase (discovery phase). Of particular importance is the dose representing the minimum anticipated concentration of biological effect (MABEL). It is an estimate of the pharmacologically active dose in humans. MABEL dose selection uses all available information from in vitro and in vivo assays, pharmacodynamic and pharmacokinetic studies, affinity of the targets, kinetics of the relationship of the receptors and the test substance, the implications on the intracellular systems and metabolic pathways, as well as rationale for selection of the species and animal model [7,8,16,17]. All of these factors are studied in the pre-clinical research stage. Potential differences in the sensitivity of the mode of action, or different signaling pathways, can have a profound impact on the results, and therefore must be taken into account and carefully evaluated.

## 5. Challenges Involved in Translating Preclinical Data to Clinical Set-Up

Many drugs that in the experimental evaluation showed a promising therapeutic effect did not have the expected pharmacodynamic effect in the patients. The possible causes are not only in the experimental context. They can also include limitations associated with the design and execution of clinical protocols [18,19], lack of correlation between the non-clinical experimental data and the protocol design, insufficient proof of concept data, and trial designs that are inconsistent with clinical endpoints. All of these aspects are highly critical in exploratory clinical trials that must answer specific questions only through human beings.

The causes attributable to experimental development have also been widely discussed in the literature [20,21], and for the case of products developed for neurodegenerative diseases, insufficient knowledge of molecular pathological mechanisms and processes represents an additional challenge.

How could the multidisciplinary approach and the regulatory approach contribute to reducing these challenges?

The regulatory guidelines have an enormous methodological value. For example, the Guideline on strategies to identify and mitigate risks for first-in-human and early clinical trials with investigational medicinal products [8] could be useful as a road map for the multidisciplinary analysis of the experimental data. It identifies all aspects that can fail in the investigational drug development process from the test substance to the design of the clinical protocol, and should guide the team’s brainstorming process.

In itself, this regulation [8] is a compendium of suggestions carried out over the past 10 years by scientists dedicated to analysing the causes of failure of clinical trials. Each point of the guideline [8] suggests where the problems and risks should be. In the multidisciplinary team during discussion, members can review each case according their specialization profile, and any possible problems could have more opportunities of detection and decision-making. The other guidelines cited [6,7,9,10] have similar value of methodological use.

The retrospective analysis of TGN 1412, taken from the Expert Scientific Group on Phase One Clinical Trials [22], can be useful to support our proposal. This scientific report presents an extensive review of non-clinical results.

It is important to bear in mind that TGN 1412’s experience contributed decisively to the development of the present regulatory body around exploratory clinical trials, microdose focus, and risk analysis. We now have the advantage of having a vision that was not possible at that time.

On 13 March 2006, six healthy volunteers received Wegener’s TGN 1412 drug that is a monoclonal antibody developed as a drug for the treatment of B cell leukemia and autoimmune diseases. Within hours of receiving TGN 1412, all six volunteers were admitted to the intensive care unit with a very severe systemic inflammatory reaction that progressed to multi-organ failure.

Cynomolgus macaque was considered a relevant species for humans because the sequence of amino acids of the loop C’’D of CD28 (target of TGN 1412) is identical to that of humans and the Rhesus monkey differs only in one amino acid.

In 2003 [23], an article was published where the incubation of T cells from healthy donors and T cells from rats were incubated with monoclonal antibodies (MABs) against human CD28, (JJ316 and JJ319) obtained through murine origin. These MABs were included in culture dishes at a final concentration of 3 μg/mL. Cells from both species showed the proliferation of T cells and the production of IL-2. The biological response was considered to be a super agonist activation. The dose calculation based on this result—performed after the clinical trial—was closer to the safety dose calculated from the clinical results. This result should have had a greater impact.

Among the reasons for not considering the rodent as a relevant species were that the amino acid sequence of CD28 between mouse and rat is 93% identical and the homology between human and mouse is only 65% [23], and differences between pharmacokinetic profiles found among non-human and rodents. Another reason that could have influenced the lack of consideration of this study could be that the monoclonal antibody (MAB) TGN1412 was not included in the study.

Subsequent studies [24] highlighted the molecular reason for the lack of pharmacodynamic response in cynomolgus macaque. They identified a species difference in CD28 expression on the CD4 effector memory T-cell subset as most likely being responsible for the failure of pre-clinical safety testing of TGN 1412 in cynomolgus macaques.

The case of TGN 1412 is a learned lesson that exposes the importance of including all experimental results in the analysis.

To this previously-indicated conclusion [22], we added others as possible strategies to reduce the risk of failures in the translation of new drugs from the experimental phase to the clinical trial:the need to establish—with biological justification—the reasons for excluding an experimental result,To introduce quality assurance criteria and regulatory focus in all phases of the research process of developing a new molecule,To use the applicable regulations as a road map to conduct the analysis of results in a multidisciplinary team context that includes specialists from technical branches related to all aspects from pharmaceutical development to clinical project design.

## 6. Concluding Remarks

As exploratory pre-clinical trials of a new drug move forward, researchers need to assume responsibility for coordinating the organization and interdisciplinary aspects discussed here; such forethought is necessary to generate high-quality and reproducible results that offer reasonable confidence for the safety of volunteers and patients who will be involved in the clinical studies based on their work. Academic and research laboratories can more effectively contribute to the process of translation from the laboratory to the patient if the important initial data they produce can be used as part of the supporting data for clinical trial authorization. Organizational changes inside and outside these institutions are required to assure logical and coherent research projects with clinical objectives: the highly detailed description of the processes, the archiving and recovery of information that allows reproducibility of the tests carried out, and the reports of compiled results that adhere to regulatory criteria are necessary components.

Effectively, this means:Introduction of quality assurance criteria for all in vitro and in vivo assays, reference and test substances and cells, characterization of candidate agents, and methods of formulation development.Appropriate facilities suitable for the performance of specific functions: type of construction, equipment with documented calibration, appropriate data storage, quality assurance system implemented.Demonstrated staff training for the specific tasks performed.

Organizational changes such as these were necessary when hospitals first assimilated Good Clinical Practices in order to participate in trials of clinical efficacy for new medications. Laboratories that wish to contribute efficiently to the development of new technologies for the diagnosis and treatment of neurodegenerative diseases must follow suit: they must incorporate criteria for quality assurance, multidisciplinary interactions, and regulatory approaches if they wish to participate in modern research strategies for new medicines that contribute to the translation of their results to the clinic.

This last admonition is the most important message from the conference, because patients suffering from neurodegenerative diseases wait with great expectation and hope for the results of scientific advances. As a bonus, introducing improved approaches to our research programs can increase opportunity and lower expenses in our laboratories.

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
