# Peer review of "Non-Clinical Contribution to Clinical Trials during Lead Optimization Phase"

_behavsci, 2018, doi:10.3390/bs8010017_

Round 1

Reviewer 1 Report

The manuscript, which is submitted under the "Conference Report", effectively outlines the guidelines related to requirements for clinical 8 trials for new drugs. Useful suggestions are made for optimizing regulatory management to improve likelihood for acceptance of pre-clinical data prior to Clinical Phase I trials. This summary is helpful in introducing improved approaches and suggesting important organizational changes. 

Author Response

The content of this paper is significant, because there are many results from basic research in neurodegenerative diseases, which is not integrated to regulatory dossier to ask authorization to perform clinical trial. Absent of relevant information from basic research could limit to do specific question during clinical trial, mistake in minimal dose calculation between other consequences.

Three message are in this paper:

·        The importance to incorporate regulatory and quality assurance criteria from basic research to give regulatory use value to these relevant data.

·        Reoriented organizational chart of labs to really incorporate to translational medicine process

·        To have a multidisciplinary team during the process of research and development, able to do entire review of data under regulatory focus, previous protocol design  of clinical trial and submission to regulatory authority.

Some challenges of translational data from experimental research to clinical proof of concept imputed to preclinical studies, could be addressed with these recommendation.

Reviewer 2 Report

Though the author suggests preclinical data could serve as potential to design clinical experiment. This review did not point out challenges involved in translating preclinical data to clinical set up. For example: lot of drugs show good response  in vivo (in mice), but fail in clinic. This shows that stage of disease, PK/PD, timing, toxicities, immunological function differs significantly.  Additionally, models used in vivo are close but do not mimic humans perfectly. Author should point the challenges and give a rational explanation of how to overcome these limitations.

Author Response

(The authors gave the same response as above.)
